# Need for manual segmentation in optical coherence tomography angiography of neovascular age-related macular degeneration

**Supriya Dabir**[1]*, **Vaidehi Bhatt**[2], **Deepak Bhatt**[3], **Mohan Rajan**[1], **Preetam Samant**[4], **Sivakumar Munusamy**[1], **C. A. B. Webers**[5], **T. T. J. M. Berendschot**[5]

**1** Department of Retina, Rajan Eye Care Pvt Ltd, Chennai, India, **2** Rajiv Gandhi Medical College, Thane, India, **3** UBM Institute, Mumbai, India, **4** Department of Retina, PD Hinduja Hospital and Medical Research Center, Mumbai, India, **5** University Eye Clinic Maastricht, Maastricht, The Netherlands

* supriad@gmail.com

**Data Availability Statement:** All relevant data are within the paper and Supporting information files.

## Abstract

### Purpose

To compare the characteristics of eyes that had manual vs. automated segmentation of choroidal neovascular membrane (CNVM) using optical coherence tomography angiography (OCTA).

### Methods

All patients with CNVM underwent OCTA using the Zeiss Angioplex Cirrus 5000. Slabs of the avascular outer retina, outer retina to choriocapillaris (ORCC) region and choriocapillaris were generated. Manual segmentation was done when there were significant segmentation artifacts. Presence of activity of CNVM was adjudged by the presence of subretinal fluid (SRF) on structural OCT and was compared to activity detected on en face OCTA slabs based on well-defined criteria.

### Results

Eighty-one eyes of 81 patients were recruited of which manual segmentation was required in 46 (57%). Eyes with automated segmentation had significantly more CNVM in the ORCC (75%) whereas those with manual segmentation had deeper CNVM (sub-RPE = 22%, intra-PED = 22%) (p<0.001). Twenty eyes (25%) were found to have active CNVM on both the structural OCT and OCTA while an additional 19 eyes were presumed to have active CNVM on OCTA alone. There was only modest concordance between disease activity detected using structural OCT and OCTA (Kappa = 0.47, 95% CI = 0.30 to 0.64).

### Conclusions

Manual segmentation of OCTA is required in more than 50% eyes with CNVM and this progressively increases with increasing depth of CNVM location from the ORCC to below the

**Funding:** Rajan eye care pvt ltd, Chennai, provided support in the form of salaries for authors (Dabir, Rajan, Sivakumar) but did not have any additional role in the study design, data collection and analysis, decision to publish, or preparation of the manuscript. The specific roles of these authors are articulated in the 'author contributions' section. This does not alter our adherence to PLOS ONE policies on sharing data and materials. Maastricht University is funding the article processing fees.

**Competing interests:** Rajan eye care pvt ltd, Chennai, provided support in the form of salaries for authors (Dabir, Rajan, Sivakumar) but did not have any additional role in the study design, data collection and analysis, decision to publish, or preparation of the manuscript. The specific roles of these authors are articulated in the 'author contributions' section. This does not alter our adherence to PLOS ONE policies on sharing data and materials.

RPE. There is moderate concordance between OCTA and structural OCT in determining CNVM activity.

## Introduction

Age related choroidal neovascular membrane (CNVM) is the leading cause of severe vision loss in the elderly [1, 2]. Ancillary testing of CNVM for disease detection as well as activity has undergone paradigm shifts over the past decade, where the non-invasive structural optical coherence tomography (OCT) has almost entirely replaced the more invasive fluorescein angiography (FFA) [2, 3]. Recent evidence shows that there is a high level of concordance in detecting disease activity between OCT and FFA at baseline, though this concordance drops off once treatment is initiated with anti-VEGF agents [4]. More recently, optical coherence tomography angiography (OCTA) that enables visualization of the different vascular layers of the retina and choroid has been extensively used to detect the presence of CNVM as well as to determine disease activity [5–8]. Both, qualitative and qualitative metrics have been developed in the assessment of OCTA images to sharpen its resemblance to disease severity and activity [9–11]. Additionally, OCTA has also enabled us to detect subclinical cases of CNVM where a neovascular complex exists without clinical signs of disease activity [12]. However, the role of OCTA in management of CNVM is still evolving.

There are many different commercially available OCTA machines based on spectral domain and swept source (SS) OCT technology [6]. Each of these uses a different proprietary algorithm to produce automated en face OCTA slabs of the superficial and deep capillary plexus in the inner retina, avascular outer retina, outer retina to choriocapillaris (ORCC) region and choriocapillaris. Most studies published on use of OCTA in CNVM utilize the automated image analysis algorithms provided by the manufacturer to determine disease characteristics [6, 13]. However, improper segmentation commonly introduces artifacts that lead to incomplete visualization of the CNVM in en face images, especially in active cases [14, 15]. Incomplete or inaccurate segmentation has wider implications in management of AMD. There is an increased likelihood of missing small early networks due to this error. Although exudation on SD-OCT in the absence of membrane on OCTA would be usually treated, but there is a distinct possibility of missing a small non-exudative membrane on OCTA due to segmentation errors. In such a scenario, the patient would be diagnosed as dry AMD and followed-up at a longer interval vis-à-vis a closer follow-up if a non-exudative membrane were detected on OCTA. Additionally, the configuration and dimensions of the network are very dependent on the section been evaluated. Automated segmentation slabs may not truly reflect the widest area of network and / or its true configuration. This can be overcome by manually readjusting the contours of the slabs to target vascular layers of special interest and generate custom en face images.

A comparison between manual and automated segmentation in eyes with active and inactive CNVM has been done in the past where Siggel et al. show that accuracy of the SS-OCTA in detecting CNVM decreases as we image deeper into the retina [16]. However, there is not enough literature studying this aspect of OCTA. In this study, we therefore compared the characteristics of eyes that had manual vs. automated analysis of CNVM using OCTA.

## Methods

This was a cross sectional observational study on consecutive consenting patients with CNVM who were referred to our center, Rajan eye care hospital Pvt Ltd, Chennai, for OCTA between

January 2018 and December 2019. The study was approved by the institutional ethics committee and was carried out as per the tenets of the declaration of Helsinki and good clinical practice guidelines. Written informed consent was obtained from all participants before enrollment.

All patients with CNVM, with or without a history of prior anti-VEGF therapy, and willing to participate in the study were recruited. The exclusion criteria were: eyes with significant sub-retinal hemorrhage and eyes with senile cataracts or poor mydriasis precluding reliable OCTA, eyes having type 3 CNVM and aneurysmal type 1 CNVM, and eyes with any other ocular comorbidity. Before performing OCTA, all patients underwent a comprehensive eye examination including evaluation of the best corrected visual acuity (BCVA), intraocular pressure (IOP), dilated slit lamp evaluation of the anterior segment and thorough evaluation of the posterior segment. Presence of CNVM was confirmed on clinical examination by a fellowship—trained retina surgeon and its presence was confirmed by visualization of the CNVM on structural OCT.

## Optical coherence tomography angiography

Patients then underwent OCTA assessment of the involved eye with CNVM using the Cirrus 5000 Zeiss Angioplex (Carl Zeiss Meditec, Dublin, CA). A well-trained technician acquired 6X6 scans centered on the fovea. Automated en face retinal angiograms of the superficial and deep retina, avascular outer retina, ORCC and choriocapillaris were created using the proprietary algorithms provided by the manufacturer. All images were interpreted and graded by an experienced clinician (DB). All OCTA images were analyzed for the need of manual vs automated segmentation by two experienced clinicians (DB and SD) A good inter-grader reliability of 0.93 was obtained.

In cases where segmentation artifacts made it difficult to visualize the entire CNVM complex, manual segmentation was done using tools in the software and custom slabs were created to visualize the CNVM in its entirety, as per protocols described by Siggel et al. [16].

Briefly, we moved the two boundaries standard segmentation pattern such that the CNVM was completely visualized. The analysis software was then utilized to analyze the image. Minimum signal strength of 7/10 was required to register the scan for analysis without any motion artifacts. Based on vascular characteristics observed on the en face OCTA, the CNVM was classified as active as per descriptions given by Coscas et al. [11]. In summary, the CNVM was deemed to be active when it was well-defined with numerous tiny branching capillaries, with anastomoses, peripheral loops and arcades at the vessel termini and/or presence of a hypo-intense halo around the CNVM (Fig 1).

An inactive CNVM was recorded when it had long and large mature linear vessels and a dead tree appearance at the vessel termini without peripheral anastomosis, loops, or arcades (Fig 2).

In addition to the OCTA, patients also underwent structural OCT in the affected eyes with CNVM (Cirrus HD OCT, Carl Zeiss Meditec, Dublin, CA). Presence of sub-retinal fluid (SRF) with or without intra-retinal fluid and cystoid spaces was considered to be active disease and this was used as gold standard for comparison of disease activity with OCTA. The same experienced examiner (DB) evaluated both the OCTA and OCT images but not simultaneously, such that he was masked for the lesion characteristics in one modality when grading disease activity in the other modality. The primary outcome measure was comparison between OCTA characteristics of eyes graded using the automated algorithm vs. eyes that required a manual segmentation. Concordance about the activity of the CNVM between OCTA and OCT was also assessed.

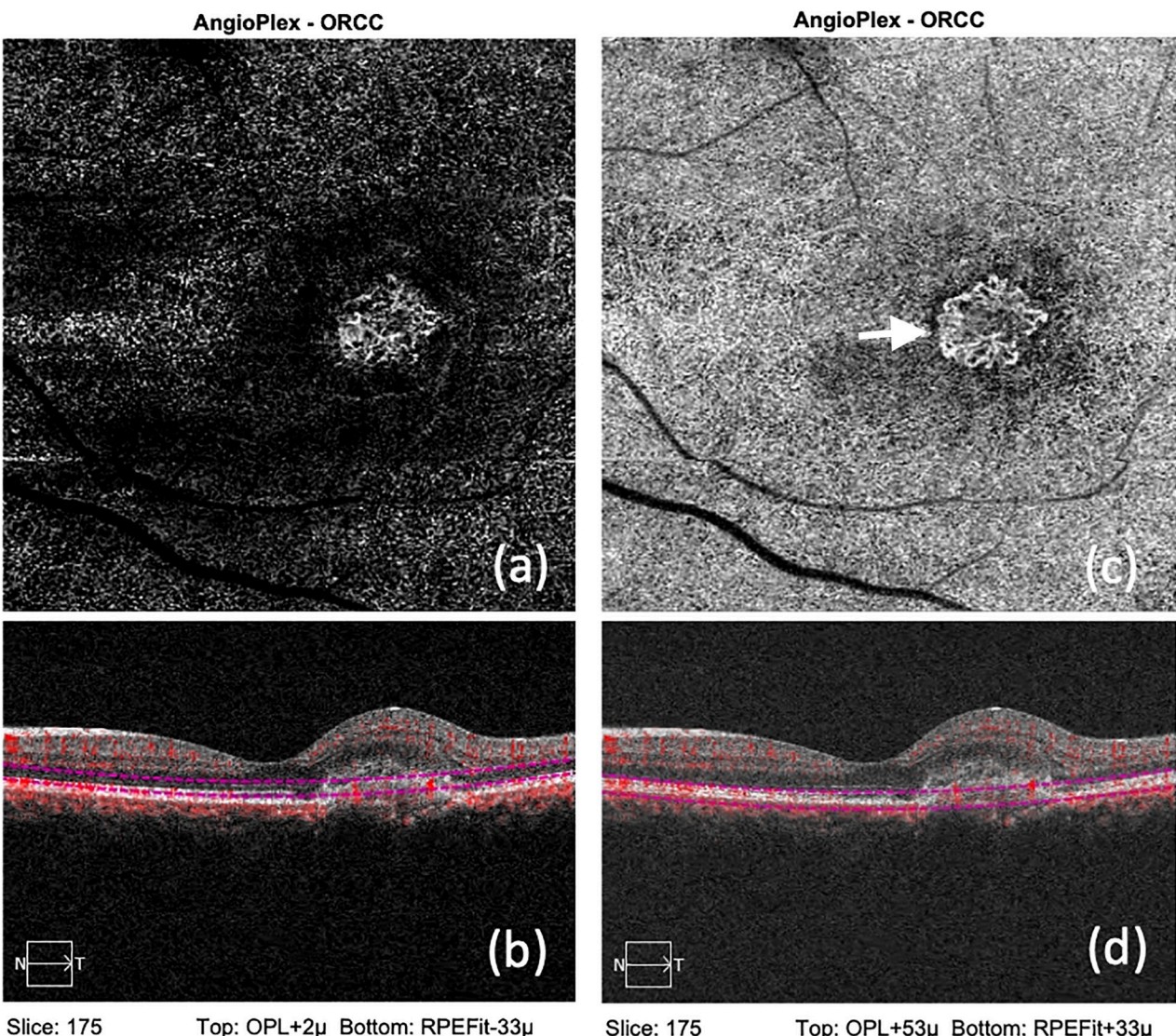

**Fig 1.** (a): Showing OCTA and structural OCT from an eye with active CNVM, with the use of automated segmentation, where the outer margins are blurred. (b): Showing OCTA and structural OCT from an eye with active CNVM, with the use of manual segmentation where the peripheral anastomoses (White arrow), which are the hallmark of activity, is well appreciated.

## Statistical analysis

All continuous variables were described as means with standard deviation and categorical variables were described as proportions (n, %). Differences between groups were analyzed using chi-square statistics and independent T-tests. The kappa statistic with 95% confidence interval (CI) was used to assess the concordance in disease activity between OCTA and OCT. All statistical analyses were done with SPSS statistical software (version 25, IBM Corp, Armonk, USA). A p-value of <0.05 was considered significant.

## Results

We included 81 eyes of 81 patients with CNVM who satisfied inclusion criteria. Automated segmentation was reliable in 35 eyes (43%) while manual segmentation was required in the

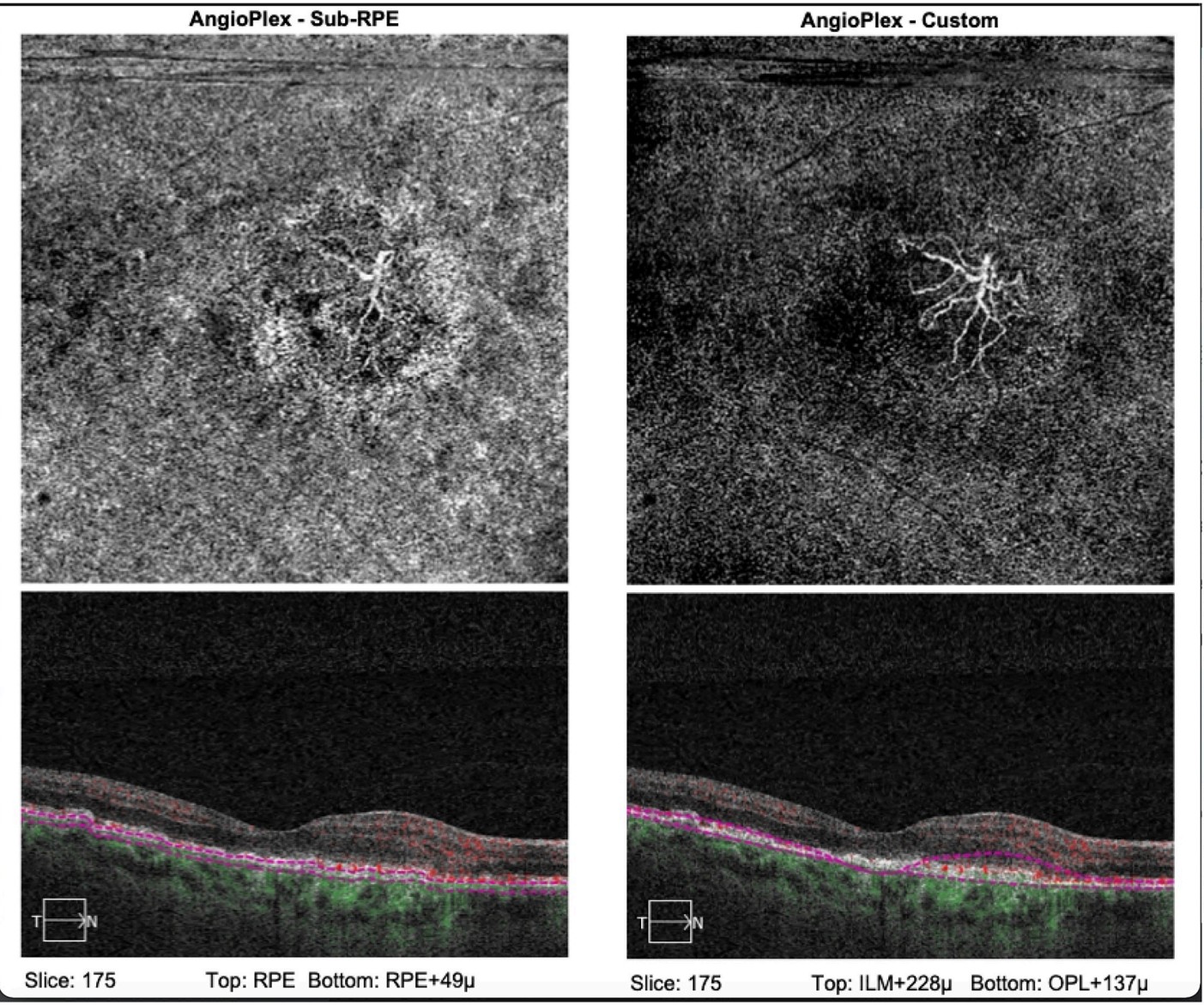

**Fig 2.** (a): Showing OCTA and structural OCT from an eye with inactive CNVM, with the use of automated segmentation, with poorly defined network. (b): Showing OCTA and structural OCT from an eye with inactive CNVM, with the use of manual segmentation where the characteristic dead-tree appearance of network, which is the hallmark of end-stage disease, is well appreciated.

remaining 46 (57%). Table 1 shows a comparison between characteristics of eyes that had manual vs. automated image analysis. Eyes that were amenable to automated analysis had the CNVM predominantly in the ORCC whereas those that needed manual segmentation had a deeper location of the CNVM, with equal numbers in the sub-RPE and intra-PED regions (Table 1). Significantly more eyes in the automated OCTA group showed presence of SRF on structural OCT, but there was no difference in number of eyes with active disease based on OCTA characteristics. Automated segmentation did not depend upon type of CNVM either (Table 1).

Out of the 81 eyes recruited, 18 (22%) had type 1 CNVM and the remaining 63 (78%) had type 2 CNVM. A comparison between eyes with type 1 and type 2 CNVM showed that

**Table 1. Descriptives for automated and manual segmentation.**

| Variable | Description | Segmentation | | |
|---|---|---|---|---|
| | | **Automated** | **Manual** | **p** |
| Number of eyes | | 35 | 46 | |
| Type of CNVM | Type 1 CNVM | 6 (17%) | 12 (26%) | 0.34 |
| | Type 2 CNVM | 29 (83%) | 34 (74%) | |
| Location | Avascular | 1 (3%) | 3 (7%) | <**0.001** |
| | Choriocapillaris | 2 (6%) | 5 (11%) | |
| | Choroidal CNVM | 0 | 2 (4%) | |
| | Deep vascular plexus | 1 (3%) | 3 (7%) | |
| | Deep plexus and Choriocapillaris | 0 | 1 (2%) | |
| | Intra-PED | 0 | **10 (22%)** | |
| | ORCC | **26 (74%)** | 8 (17%) | |
| | RPE-RPE Fit | 0 | 4 (8%) | |
| | SUB-RPE | 5 (14%) | **10 (22%)** | |
| SRF | Yes | **13 (37%)** | **7 (15%)** | **0.023** |
| Activity | Yes | 20 (57%) | 19 (41%) | 0.16 |
| Halo around CNVM | Yes | 5 (14%) | 7 (15%) | 0.74 |
| Peripheral arcades | Yes | 14 (40%) | 9 (20%) | 0.073 |
| Terminal Anastomosis | Yes | 12 (34%) | 11 (24%) | 0.46 |

**Table 2. Activity on OCTA versus activity on structural OCT.**

| | | Activity on OCTA | | Total |
|---|---|---|---|---|
| | | **Yes** | **No** | |
| Activity on structural OCT | Yes | 19 (49%) | 1 (2%) | 20 (25%) |
| | No | 20 (51%) | 41 (98%) | 61 (75%) |
| Total | | 39 (100%) | 42 (100%) | 81 (100%) |

significantly more eyes in the type 1 group (n = 10, 56%) had presence of SRF on structural OCT compared to type 2 eyes (n = 10, 16%) (p<0.001). Eyes with type 1 disease had marginally higher proportion of eyes detected as active on OCTA (n = 12, 67%) compared to type 2 disease (n = 27, 43%) (p = 0.075).

Of the total of 81 eyes, 20 eyes (25%) were found to have active CNVM on the structural OCT on account of SRF whereas 39 eyes (47%) were found to harbor active CNVM on OCTA. On cross tabulating disease activity detected on OCT and OCTA (Table 2), we found that almost all eyes with activity on OCT showed active CNVM on OCTA. However, there was only modest concordance between disease activity detected using OCT and OCTA (Kappa = 0.47, 95% CI = 0.30 to 0.64) with half the eyes showing active CNVM on OCTA demonstrating no signs of disease activity on structural OCT. On OCTA, the hypo-reflective halo around the CNVM was the least common finding in active lesions, both in the automated (25%) and manual (32%) segmentation groups.

## Discussion

We found that a majority of eyes with CNVM that could be successfully analyzed using automated segmentation had the CNVM complex in the ORCC whereas more than half the eyes that required manual segmentation had a deeper CNVM complex, i.e. either sub-RPE or intra-PED or in the choriocapillaris. Additionally, we noted that only half the eyes deemed to have

active CNVM on OCTA demonstrated signs of disease activity on structural OCT. Branching pattern with peripheral loops, anastomosis and arcades were the main findings that lead us to label a CNVM as active while halo around the CNVM was the least helpful finding seen in only a third of the eyes. A significantly greater number of eyes with type 1 CNVM had SRF compared to type 2 eyes.

Ever since the advent of OCTA for evaluation of retinal and choroidal vascular diseases nearly a decade ago, it has been applied to study the characteristics of CNVM, especially secondary to CNVM [2, 17, 18]. Coscas et al. described the qualitative features of CNVM in terms of the branching pattern to determine disease activity [11]. We used the same features to determine activity in our study. However, most studies have used the automated segmentation available with the OCTA machines to create the vascular slabs, though this may not always be feasible due to segmentation artifacts that make OCTA interpretation difficult and inaccurate [14, 15]. We found that automated segmentation provided by the machine was reliable in less than half the cases in our series. In an excellent study published recently, Siggel et al. studied 102 eyes to determine the sensitivity and specificity of automated versus manually segmented OCTA images to detect presence of CNVM [16]. They showed that automated slabs at the ORCC had the highest ability to detect CNVM compared to the gold standard FFA, while slabs at the choriocapillaris level had the lowest. However, sensitivity of detection could be increased substantially using manual segmentation to produce custom slabs. They also found a significantly higher concordance between FFA and OCTA in detecting CNVM when manual segmentation was used as opposed to only moderate concordance when automated segmentation was used. Unfortunately, they did not comment on CNVM activity. Yet, our data coupled with theirs clearly shows that deeper CNVM complexes below the RPE require manual segmentation much more often than complexes in the ORCC.

Manual segmentation requires a lot of effort and time, has a steep learning curve and hence may be difficult to perform in high volume retina clinics where time is at a premium. Lauermann et al. have shown that segmentation artifacts occur most often in CNVM eyes [15]. However, when structural OCT shows a sub-RPE CNVM complex, interpreting the OCTA images from automated segmentation should be done with caution and should be relied upon only when the entire CNVM complex is visualized without any segmentation or motion artifacts. Retina specialists should be aware that manual segmentation would be needed on nearly half the occasions and hence additional time should be budgeted for this. Since OCTA utilizes differences in the phase and intensity information contained within sequential B-scans performed at the same position, it is possible that the RPE, especially in diseased states, interferes with signal phase and intensity transmission, making automated segmentation unreliable.

We were surprised to find that only half the eyes that were labeled as active CNVM on OCTA actually showed signs of disease activity on the structural OCT. A recent post-hoc analysis from the HARBOR study showed that the concordance between FFA and OCT in detecting disease activity was 99% in treatment naïve eyes with CNVM while this dropped to 36% at 2 years following treatment with anti-VEGF agents [4]. Considering our dataset, where there were many eyes with inactive disease, possibly due to varying levels of anti-VEGF exposure, we believe that structural OCT was unable to pick up disease activity whereas OCTA demonstrated this better. If this were indeed true, then eyes where treatment was withheld due to no activity on OCT but activity on OCTA, would develop signs of activity on OCT subsequently and require retreatment. This is what is seen in real-world clinical practice, where lack of a reliable structural indicator of CNVM activity on OCT has led us to adopt the treat and extent treatment regimen. With OCTA giving us a potentially better indication of disease activity, it may be prudent to adopt it for determining treatment protocols for individual patients, though our study is not adequately powered to make robust recommendations. On similar lines, a

recently published by Corvi et al found that OCTA appears to be superior to other imaging modalities (FFA, ICGA and OCTA) for identification of CNVM in eyes with macular atrophy [19]. However, OCTA is still evolving and more data is required before we use it ahead of structural OCT to determine disease activity.

This study has a couple of limitations, namely the lack of accurate history of anti-VEGF treatment and lack of FFA to compare disease activity. Additionally, the smaller number of eyes with type 1 disease, which is seen more often in clinical practice, makes it difficult to make generalized recommendations, although we feel that the trends may be applicable to most cases. On the other hand, the advantages of this study are the masking of the grader to outputs from other modalities at the time of grading, use of standardized and well-established definitions of CNVM activity on OCTA and use of an experienced grader capable of performing manual segmentation accurately when indicated.

In conclusion, OCTA requires manual segmentation to generate custom slabs and visualize the vasculature of interest whenever artifacts arise from automated segmentation. Need for manual segmentation progressively increases from CNVM in the ORCC to below the RPE and choriocapillaris. There is only a moderate concordance between OCTA and structural OCT in determining CNVM activity at present.

## Supporting information

**S1 File. Minimal anonymized data.**
(XLSX)

## Author Contributions

**Conceptualization:** Supriya Dabir, Deepak Bhatt.

**Data curation:** Supriya Dabir, Vaidehi Bhatt, Deepak Bhatt.

**Formal analysis:** Deepak Bhatt, T. T. J. M. Berendschot.

**Investigation:** Supriya Dabir, Vaidehi Bhatt, Mohan Rajan, Preetam Samant, Sivakumar Munusamy.

**Methodology:** Deepak Bhatt, Mohan Rajan, T. T. J. M. Berendschot.

**Resources:** Deepak Bhatt.

**Software:** Vaidehi Bhatt, Deepak Bhatt, Sivakumar Munusamy, T. T. J. M. Berendschot.

**Supervision:** Mohan Rajan, C. A. B. Webers.

**Validation:** Deepak Bhatt, Mohan Rajan, Preetam Samant, C. A. B. Webers.

**Writing – original draft:** Supriya Dabir.

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
