## [Decision Letter · Decision Letter 0]

11 Nov 2020

PONE-D-20-31372

Need for manual segmentation in optical coherence tomography angiography of neovascular age-related macular degeneration

PLOS ONE

Dear Dr. Dabir,

Thank you for submitting your manuscript to PLOS ONE. After careful consideration, we feel that it has merit but does not fully meet PLOS ONE’s publication criteria as it currently stands. Therefore, we invite you to submit a revised version of the manuscript that addresses the points raised during the review process.

Please change the abbreviation MNV to CNV, which is the conventional usage.  Two graders are needed to evaluate the lesions using OCTA, and present the agreement between graders..

We look forward to receiving your revised manuscript.

Kind regards,

Alfred S Lewin, Ph.D.

Academic Editor

PLOS ONE

Journal Requirements:

3.We note that you have indicated that data from this study are available upon request. PLOS only allows data to be available upon request if there are legal or ethical restrictions on sharing data publicly. For more information on unacceptable data access restrictions, please see http://journals.plos.org/plosone/s/data-availability#loc-unacceptable-data-access-restrictions.

4.Thank you for stating the following in the Financial Disclosure section:

[The author(s) received no specific funding for this work.].   

We note that one or more of the authors are employed by a commercial company: Rajan Eye Care Pvt Ltd

Reviewers' comments:

Reviewer's Responses to Questions

**Comments to the Author**

1. Is the manuscript technically sound, and do the data support the conclusions?

Reviewer #1: Partly

Reviewer #2: Partly

2. Has the statistical analysis been performed appropriately and rigorously? 

Reviewer #1: Yes

Reviewer #2: Yes

3. Have the authors made all data underlying the findings in their manuscript fully available?

Reviewer #1: No

Reviewer #2: Yes

4. Is the manuscript presented in an intelligible fashion and written in standard English?

Reviewer #1: Yes

Reviewer #2: Yes

5. Review Comments to the Author

Reviewer #1: 1. The manuscript is interesting.

2. The abbreviation MNV may be replaced by CNV, which is more conventionally used.

3. Please mark the ‘peripheral anastomoses’ in figure 1b,

4. What is the significance of figure 2, showing 3 different eyes? Rather, it would add value to present images of the case examples exhibiting statistical difference between automated and manual segmentation. For example, cases where automated slabs failed to show deeper CNV, while manual segmentation detected the deeper CNVs

Reviewer #2: Dear Editor

Thanks very much for asking me to review this study entitled “Need for manual segmentation in optical coherence tomography angiography of neovascular age-related macular degeneration”

The study is interesting but I have some concern about several points:

- I suggest to not start the paper with “Age related macular neovascularization (MNV)” it is weird.

- Introduction: “More recently, optical coherence tomography angiography (OCTA) that enables visualization of the different vascular layers of the retina and choroid has been extensively used to detect the presence of MNV as well as to determine disease activity.5,6” I suggest to reference also: 1) Comparison of SD-Optical Coherence Tomography Angiography and Indocyanine Green Angiography in Type 1 and 2 Neovascular Age-related Macular Degeneration (doi: 10.1167/iovs.17-22902) and 2) Comparison between several optical coherence tomography angiography devices and indocyanine green angiography of choroidal neovascularization (doi: 10.1097/IAE.0000000000002471.)

- In the introduction the purpose of the authors should be explained better. I mean, it is hard to understand the reason of this study.

- Consecutive eyes with MNV were included. What about Type 3 MNV or aneurysmal Type 1 MNV?

- The disease activity based on OCTA has no reason here. It cannot be evaluated in this way. The definition of active and inactive MNV based on the appearance of the vessels it is not supported. There are some MNV with large mature linear vessels and a dead tree appearance and subretinal fluid, does it mean that is not active? The appearance of the vessels is more related to the “age” of the lesion rather than the activity. I suggest to remove all the part related to the activity evaluation. Moreover, this is not a prospective study but a cross-sectional study. It is a bet to assess the activity of the lesion in this way and it has no reason to be presented.

- Two graders are needed to evaluate the lesions using OCTA. And agreement between graders should be presented.

6. PLOS authors have the option to publish the peer review history of their article (what does this mean?). If published, this will include your full peer review and any attached files.

Reviewer #1: No

Reviewer #2: No

---

## [Author Response · Author response to Decision Letter 0]

6 Dec 2020

Reply to the reviewers’ comments

Reviewer Number Original comments of the reviewer Reply by the author(s)

Journal Requirements 1 Please ensure that your manuscript meets PLOS ONE's style requirements, including those for file naming. We have made the necessary changes.

Journal Requirements 2 Please provide additional details regarding participant consent. In the ethics statement in the Methods and online submission information, please ensure that you have specified what type you obtained (for instance, written or verbal, and if verbal, how it was documented and witnessed). If your study included minors, state whether you obtained consent from parents or guardians. If the need for consent was waived by the ethics committee, please include this information.

 Written informed consent was obtained from all the participants which has been included in the Methods section of manuscript. 

Journal Requirements 3 We note that you have indicated that data from this study are available upon request. PLOS only allows data to be available upon request if there are legal or ethical restrictions on sharing data publicly. For more information on unacceptable data access restrictions, please see http://journals.plos.org/plosone/s/data-availability#loc-unacceptable-data-access-restrictions.

We will update your Data Availability statement on your behalf to reflect the information you provide. We do not have any restriction on sharing the dataset and have upload the minimal anonymized dataset. The same has been mentioned in the cover letter too.

Journal Requirements 4 Thank you for stating the following in the Financial Disclosure section:

[The author(s) received no specific funding for this work.]. 

We note that one or more of the authors are employed by a commercial company: Rajan Eye Care Pvt Ltd

Please know it is PLOS ONE policy for corresponding authors to declare, on behalf of all authors, all potential competing interests for the purposes of transparency. PLOS defines a competing interest as anything that interferes with, or could reasonably be perceived as interfering with, the full and objective presentation, peer review, editorial decision-making, or publication of research or non-research articles submitted to one of the journals. Competing interests can be financial or non-financial, professional, or personal. Competing interests can arise in relationship to an organization or another person. Please follow this link to our website for more details on competing interests: http://journals.plos.org/plosone/s/competing-interests.

 We have made the necessary changes in Funding Statement and Competing Interests Statement, and the same have also been included in the Cover letter. 

Journal Requirements 5 Your ethics statement should only appear in the Methods section of your manuscript. If your ethics statement is written in any section besides the Methods, please delete it from any other section. We have made the necessary changes.

Reviewer #1: 1 The manuscript is interesting. Thank you for your comments.

Reviewer #1: 2 The abbreviation MNV may be replaced by CNV, which is more conventionally used. We agree with the reviewers’ suggestion. We have made the necessary changes.

Reviewer #1: 3 3. Please mark the ‘peripheral anastomoses’ in figure 1b, The marking of ‘peripheral anastomoses’ has been included in the modified figure 1b. 

Reviewer #1: 4 What is the significance of figure 2, showing 3 different eyes? Rather, it would add value to present images of the case examples exhibiting statistical difference between automated and manual segmentation. For example, cases where automated slabs failed to show deeper CNV, while manual segmentation detected the deeper CNVs

 We agree with the reviewers’ suggestion. We have provided the modified image as requested. 

Reviewer #2: 1 I suggest to not start the paper with “Age related macular neovascularization (MNV)” it is weird.

 We agree with the reviewers’ suggestion. We have made the necessary changes. 

Reviewer #2: 2 

Introduction: “More recently, optical coherence tomography angiography (OCTA) that enables visualization of the different vascular layers of the retina and choroid has been extensively used to detect the presence of MNV as well as to determine disease activity.5,6” I suggest to reference also: 1) Comparison of SD-Optical Coherence Tomography Angiography and Indocyanine Green Angiography in Type 1 and 2 Neovascular Age-related Macular Degeneration (doi: 10.1167/iovs.17-22902) and 2) Comparison between several optical coherence tomography angiography devices and indocyanine green angiography of choroidal neovascularization (doi: 10.1097/IAE.0000000000002471.)

Thank you for this important suggestion. We have included both the references in our manuscript. 

Reviewer #2: 3 

In the introduction the purpose of the authors should be explained better. I mean, it is hard to understand the reason of this study.

Thank you for the suggestion. We have modified the introduction accordingly. 

Reviewer #2: 4 Consecutive eyes with MNV were included. What about Type 3 MNV or aneurysmal Type 1 MNV?

 In our study, we have excluded eyes with type 3 CNVM and aneurysmal type 1 CNVM. We have added this into the Methods. 

Reviewer #2: 5 The disease activity based on OCTA has no reason here. It cannot be evaluated in this way. The definition of active and inactive MNV based on the appearance of the vessels it is not supported. There are some MNV with large mature linear vessels and a dead tree appearance and subretinal fluid, does it mean that is not active? The appearance of the vessels is more related to the “age” of the lesion rather than the activity. I suggest to remove all the part related to the activity evaluation. Moreover, this is not a prospective study but a cross-sectional study. It is a bet to assess the activity of the lesion in this way and it has no reason to be presented.

 Thank you for raising this pertinent point. Segmentation forms a very important role in delineating the network in AMD. Due to segmentation errors, there is an increased likelihood of missing small early networks. Additionally, the configuration and dimensions of the network are very dependent on the section been evaluated. Automated segmentation slabs may not truly reflect the widest area of network and / or its true configuration and /or if it is active based on criteria such as presence of tiny branching capillaries, with anastomoses, peripheral loops and arcades at the vessel termini and/or presence of a hypo-intense halo around the CNVM. These finer features of active network can be missed if the automated segmentation slab fails to detect them, and the network would be wrongly labelled as inactive. This can be overcome by manually readjusting the contours of the slabs to target vascular layers of special interest and truly define its characteristics. Since our aim was to compare between manual and automated segmentation, evaluation of disease activity on OCTA inevitably became an integral part of the study. 

With regards to defining disease activity on OCTA, Coscas et al in their landmark paper on OCTA in AMD have concluded that “The presence of tiny branching vessels and a peripheral anastomotic arcade appears to predict the lesion activity with a good accuracy and the model based on four criteria enables optimal decisions regarding retreatment in eAMD.” [Coscas F, Lupidi M, Boulet JF, et al. Optical coherence tomography angiography in exudative age-related macular degeneration: a predictive model for treatment decisions. British Journal of Ophthalmology 2019;103:1342-1346.] The additional two OCTA criteria in their study which defined disease activity were “vascular loops” and “choriocapillaris hypointense”. Based on this landmark paper, we have defined AMD disease activity on OCTA as “presence of numerous tiny branching capillaries, with anastomoses, peripheral loops and arcades at the vessel termini and/or presence of a hypo-intense halo around the CNVM”. Darwish et al too have utilized these criteria for disease activity and classified AMD into 2 patterns, those requiring treatment (Pattern 1) and those not requiring treatment (Pattern 2) [Darwish A. OCT angiography for the evaluation of wet age related macular degeneration. EC Ophthalmol.2017;6:06-18.]. They too concluded that “OCT angiography may be clinically useful to evaluate the CNV activity and response to treatment as well as to differentiate the various types of CNV in wet AMD.” 

Although we agree with the reviewer that defining disease activity on OCTA does have its own limitations, including a scenario where the patient may have subretinal fluid but mature vessels on OCTA. However, in such a scenario it is important to point out that “presence of SRF” is an “SD-OCT biomarker of disease activity” and intermingling it with “OCTA disease activity” was never the intention of our study. Our case definition of disease activity on both SD-OCT and OCTA is very precise and independent of each other. In fact, Lek JJ et al have described a new entity of nonexudative detachment of the neurosensory retina (NEDNR), in which there is presence of subclinical SRF in the absence of evidence of neovascularization or polyp [Lek JJ, Caruso E, Baglin EK, Sharangan P, Hodgson LAB, Harper CA, Rosenfeld PJ, Luu CD, Guymer RH. Interpretation of Subretinal Fluid Using OCT in Intermediate Age-Related Macular Degeneration. Ophthalmol Retina. 2018 Aug;2(8):792-802.]. Thus, presence of SRF on OCT does not necessary imply presence of network or even disease activity. In addition, a small amount can very well be tolerated in nAMD as demonstrated by the FLUID study [Guymer RH, Markey CM, McAllister IL, Gillies MC, Hunyor AP, Arnold JJ; FLUID Investigators. Tolerating Subretinal Fluid in Neovascular Age-Related Macular Degeneration Treated with Ranibizumab Using a Treat-and-Extend Regimen: FLUID Study 24-Month Results. Ophthalmology. 2019 May;126(5):723-734.]. Furthermore, they even suggested that “it is possible

that a subretinal space may exist long after active

exudation has ceased, with failure of the retina to adhere

firmly back to the retinal pigment epithelium through

interdigitations.”

In summary, our definition of disease activity and associated biomarkers on SD-OCT and OCTA was based upon the existing body of literature as elucidated above. We hope to get further insights into disease activity aspect on OCTA in nAMD once the results of the prospective “Defining Disease Activity in Neovascular AMD With Optical Coherence Tomography Angiography (DANA)” study are published. (https://clinicaltrials.gov/ct2/show/record/NCT03909425?view=record ).

Reviewer #2: 6 Two graders are needed to evaluate the lesions using OCTA. And agreement between graders should be presented. Thank you for this observation. Although two graders had evaluated the lesion, this data was not provided in the original manuscript. Based on reviewer’s suggestion, we have calculated the intergrader agreement, which was very reliable (0.93). The same has been added in the Methods and the data has also been provided into the minimal anonymized dataset.

---

## [Decision Letter · Decision Letter 1]

17 Dec 2020

Need for manual segmentation in optical coherence tomography angiography of neovascular age-related macular degeneration

PONE-D-20-31372R1

Dear Dr. Dabir,

We’re pleased to inform you that your manuscript has been judged scientifically suitable for publication and will be formally accepted for publication once it meets all outstanding technical requirements.

Kind regards,

Alfred S Lewin, Ph.D.

Section Editor

PLOS ONE

Additional Editor Comments (optional):

Reviewers' comments:

Reviewer's Responses to Questions

**Comments to the Author**

1. If the authors have adequately addressed your comments raised in a previous round of review and you feel that this manuscript is now acceptable for publication, you may indicate that here to bypass the “Comments to the Author” section, enter your conflict of interest statement in the “Confidential to Editor” section, and submit your "Accept" recommendation.

Reviewer #1: All comments have been addressed

Reviewer #2: All comments have been addressed

2. Is the manuscript technically sound, and do the data support the conclusions?

Reviewer #1: No

Reviewer #2: Yes

3. Has the statistical analysis been performed appropriately and rigorously? 

Reviewer #1: Yes

Reviewer #2: Yes

4. Have the authors made all data underlying the findings in their manuscript fully available?

Reviewer #1: No

Reviewer #2: Yes

5. Is the manuscript presented in an intelligible fashion and written in standard English?

Reviewer #1: Yes

Reviewer #2: Yes

6. Review Comments to the Author

Reviewer #1: The track changes should have been removed from the submitted manuscript. Otherwise the comments have been addressed.

Reviewer #2: Dear Editor,

Thank you very much for asking me to re-evaluate this interesting paper.

The issues raised in the previous submission have been favorably taken in account by the authors. The presentation of the data is more effective.

7. PLOS authors have the option to publish the peer review history of their article (what does this mean?). If published, this will include your full peer review and any attached files.

Reviewer #1: No

Reviewer #2: No

---

## [Editor Report · Acceptance letter]

22 Dec 2020

PONE-D-20-31372R1 

Need for manual segmentation in optical coherence tomography angiography of neovascular age-related macular degeneration 

Dear Dr. Dabir:

I'm pleased to inform you that your manuscript has been deemed suitable for publication in PLOS ONE. Congratulations! Your manuscript is now with our production department. 

Kind regards, 

on behalf of

Dr. Alfred S Lewin 

Section Editor

PLOS ONE